# Multilocus Data Analysis Reveal the Diversity of Cryptic Species in the *Tillandsia ionantha* (Bromeliaceae: Tillansiodeae) Complex

**DOI:** 10.3390/plants11131706

**Published:** 2022-06-28

**Authors:** Juan J. Ancona, Juan P. Pinzón-Esquivel, Eduardo Ruiz-Sánchez, Clarisse Palma-Silva, Juan J. Ortiz-Díaz, Juan Tun-Garrido, Germán Carnevali, Néstor E. Raigoza

**Affiliations:** 1Departamento de Botánica-Herbario UADY, Campus de Ciencias Biológicas y Agropecuarias, Universidad Autónoma de Yucatán, Carretera Mérida-Xmatkuil km 15.5, Mérida 97315, Mexico; juan.pinzone@correos.uady.mx (J.P.P.-E.); odiaz@correo.uady.mx (J.J.O.-D.); tgarrido@correo.uady.mx (J.T.-G.); 2Departamento de Botánica y Zoología, Centro Universitario de Ciencias Biológicas y Agropecuarias, Universidad de Guadalajara, Las Agujas, Zapopan 45200, Mexico; ruizsanchez.eduardo@gmail.com; 3Departamento de Biologia Vegetal, Universidade Estadual de Campinas, Rua Monteiro Lobato 255, CEP, Campinas 13083-862, Brazil; cpalma@unicamp.com; 4Unidad de Recursos Naturales, Centro de Investigación Científica de Yucatán A. C. Calle 43 #130, Colonia Chuburná de Hidalgo, Mérida 97215, Mexico; carneval@cicy.mx (G.C.); nestor.raigoza@cicy.mx (N.E.R.)

**Keywords:** allopatric speciation, coalescent species delimitation, cryptic species, *Tillandsia ramireziana*, *Tillandsia vanhyningii*

## Abstract

Independent evolutionary lineages or species that lack phenotypic variation as an operative criterion for their delimitation are known as cryptic species. However, these have been delimited using other data sources and analysis. The aims of this study are: (1) to evaluate the divergence of the populations of the *T. ionantha* complex; and (2) to delimit the species using multilocus data, phylogenetic analysis and the coalescent model. Phylogenetic analyses, genetic diversity and population structure, and isolation by distance analysis were performed. A multispecies coalescent analysis to delimit the species was conducted. Phylogenetic analysis showed that *T. ionantha* is polyphyletic composed of eight evolutionary lineages. Haplotype distribution and genetic differentiation analysis detected strong population structure and high values of genetic differentiation among populations. The positive correlation between genetic differences with geographic distance indicate that the populations are evolving under the model of isolation by distance. The coalescent multispecies analysis performed with starBEAST supports the recognition of eight lineages as different species. Only three out of the eight species have morphological characters good enough to recognize them as different species, while five of them are cryptic species. *Tillandsia scaposa* and *T. vanhyningii* are corroborated as independent lineages, and *T. ionantha* var. *stricta* changed status to the species level.

## 1. Introduction

Cryptic species refers to taxa that are erroneously lumped into a single nominal species due to the paucity of conspicuous morphological differences, but have been delimited from other sources of data and analysis [1,2,3]. Historically, morphological criteria have been used to identify and delimit species. However, relying solely on morphology will likely result in an underestimation of the number of species [4]. This fact has required the use of other data sources (e.g., molecular, ecological, and biogeographical) and analytical methods (e.g., phylogeographical, population genetics, phylogenetic, and coalescent based) for the recognition of cryptic lineages [2,5,6,7].

The emergence of molecular and genomic datasets, as well as contemporary species concepts such as the one that considers species as independent evolutionary lineages [8,9], have brought species delimitation to an interesting crossroads with various methodological and philosophical approaches [10]. Currently, several effective techniques and methods have been developed for the more accurate delimitation and classification of species; among these are phylogenetic and non-phylogenetic methods [11] as well as Bayesian methods based on the coalescent model [12,13,14]. Recently, the application of the coalescent theory provides a fundamentally different and stronger framework to objectively identify cryptic and/or allopatric species using genetic data [10]. In addition, coalescent methods allow us to infer dynamics of divergence, interaction of evolutionary processes and relationships among taxa [15].

In this study, our model taxa are the so-called *Tillandsia ionantha* Planch. complex. Ancona et al. [16] postulated that the *T. ionantha* complex is composed of two species, five varieties, and two forms: *Tillandsia scaposa* (L.B. Sm.) Ehlers and *T. ionantha* Planch. (*T. ionantha* var*. maxima* Ehlers, *T. ionantha* var*. ionantha*, *T. ionantha* var*. stricta* f. *fastigiata* Koide, *T. ionantha* var*. stricta* Koide f*. stricta*, *T. ionantha* var. *vanhyningii* M.B. Foster and *T. ionantha* var. *zebrina* B.T. Foster). The complex is widely distributed in the tropical dry forests, savannahs, and oak forests in transition with the dry forests of Mexico and Central America, and its elevation ranges from 60 to 1600 m. Morphologically, the members of the *T. ionantha* complex are characterized by having relatively small rossetes compared to others members of the subgenus *Tillandsia* (20 cm length). In general, they are epiphytic and occasionally rupicolous. They present 10 (–20) cm tall rosettes, acaulescent or caulescent, with green or greyish leaves that turn red at anthesis and that are covered by a silvery-lepidote coating; they have 2–3 (4) flowers nested in the center, with violet petals bearing exerted stamens and stigmas [17].

*Tillandsia scaposa*, was originally described as a variety of *T. ionantha* (=*T. ionantha* var. *scaposa* L.B.Sm.). Ehlers [18] nowadays treated as a different species due to the presence of a pedunculated inflorescence. *Tillandsia ionantha* var. *vanhyningii* was described by Foster [19]. At first, this species was considered as a new species, but when compared with *T. ionantha*, no differences were found other than the rupicolous habit and caulescent rosettes, but the arrangement and position of the flowers, color of the petals and exserted stamens and pistil led him to conclude that it was simply a variant. After the publication of Ancona et al. [16], Beutelspacher and García-Martínez [20] changed the status of *T. ionantha* var. *vanhyningii* to species. Other varieties and forms described, do not present morphological variation in reproductive structures; the descriptions were made based on vegetative morphological characters, which are highly variable between and within wild populations [16,21]. However, among these taxa there are populations that are isolated by geographical barriers that might play an important role in allopatric speciation.

The phylogenetic position of the *Tillandsia ionantha* complex has been evaluated using molecular data [22]. With four cpDNA markers, the results showed that *T. ionantha* in its current constituency is polyphyletic. Two large clades were also detected: one integrated by the populations of the Pacific slope and another by the slopes of the Gulf of Mexico and Central America; in addition to *T. scaposa* and *T. ionantha* var. *vanhyningii* form, two clades isolated from the rest of the varieties of *T. ionantha* [22]. It is important to highlight that Rodríguez’s objective was to determine the phylogenetic position of the *T. ionantha* complex within the subgenus *Tillandsia* and not to delimit species in the complex. But his results suggest lineage hypotheses that should be tested. Thus, in this study, we are interested in answering the following questions: Are *Tillandsia scaposa* and *T. vanhyningii* independent species? Is there population genetic structure and genetic isolation among populations? If such a divergence exists, is geographic distance a factor in the evolution of *T. ionantha* populations? Are cryptic species present in the *T. ionantha* complex? Therefore, the aims of this study are (1) to evaluate the genetic variation between the populations and varieties of the *T. ionantha* complex, and (2) to delimit its species using multiple genomic regions and phylogenetic analyses.

## 2. Results

### 2.1. Phylogenetic Analysis

The topology of the phylogenetic analysis with Bayesian inference and Maximum Likelihood (ML) were identical in both sets of markers. Here we only present the phylogenetic trees of Bayesian inference for cpDNA and nDNA markers. On the branches, we feature the bootstrap support values resulting from the phylogenetic analysis based upon ML. The phylogenetic tree obtained from the analysis of three cpDNA markers showed that *Tillandsia ionantha* in its current circumscription is polyphyletic (Figure 1). In general, the complex is arranged in three major clades: (1) a clade that includes the *T. ionantha* lineage 1, *T. ionantha* var. *stricta*, *T. scaposa*, *T. ionantha* lineage 2 plus the species *T. strepotphylla* Scheidw. ex E. Morren, *T. pruinosa* Sw. and *T. seleriana* Mez. (2) An intermediate clade that includes the lineage 3 and lineage 4. These populations are phylogenetically isolated from the rest of the varieties of *T. ionanatha* and more closely related to the species of the outgroup from *T. lydiae* Ehlers to *T*. *variabilis* Schltdl. (3) Finally, a clade that includes *T. ionantha* lineage 5 and *T. dasyliriifolia* Baker as a sister species. Other outgroup species plus *T. vanhyningii* are inserted into this clade.

Within the complex, eight evolutionary lineages with high support values are observed. The populations of *Tillandsia ionantha* var. *stricta*, *T. vanhyningii*, and *T. scaposa* are recovered as three monophyletic lineages, whereas populations of *T. ionantha* var. *ionantha* segregated into five lineages. Lineage 1 is made up of the Central American populations and includes the type locality of *T. ionantha* var. *zebrina* (Nicaragua, Guastatoya and Zacapa). Lineage 2 is made up of the populations of *T. ionantha* var. *ionantha* from northeastern Mexico (Jalcomulco, El Higo, Tolimán, and El Troncón). Lineages 3 and 4 comprise the populations of *T. ionantha* var. *ionantha* from Pochotitan and Matatán, respectively. Lineage 5 is composed of the populations of *T. ionantha* var. *ionantha* from the Pacific lowlands and Lower Balsas Basin, including the type locality of *T. ionantha* var. *maxima* (Jalisco, Pilcaya, Santo Tomás de los Plátanos, el Puente, and Huamelula).

Phylogenetic analysis with Bayesian Inference using the *PHYC* marker array shows a different typology than that obtained with cpDNA markers. With this nuclear marker, *T. ionantha* in its current district is paraphyleticand sister to *T. scaposa*. However, within the complex, the eight lineages recognized with the cpDNA markers are recovered with high values of posterior probability and bootstrap (Figure 2). *Tillandsia ionantha* lineage 5 and *T. vanhyniningii* are sister lineages.

### 2.2. Genetic Diversity and Structure of cpDNA

The alignment was 3367 bp long and consisted of 229 nDNA sequences, with 45 polymorphic sites resulting in 13 haplotypes. The estimated haplotype diversity was *Hd* = 0.83 and the nucleotide diversity π = 0.0025. Genetic diversity across all populations (*hT* = 0.888; *vT* = 0.884) was greater than within population average value (*hS* = 0.076; *vS* = 0.009). The 13 haplotypes identified are strongly associated with biogeographic districts representing the same pattern observed with phylogenetic analyses (Figure 3A,B). The Oaxacan Plateu biogeographic district featured the highest number of haplotypes (haplotypes 8, 9, 10, and 11), followed by the Sinaloan district with three haplotypes (haplotypes 5, 6 and 7). On the other hand, haplotype 4 is the one with the widest geographical distribution and it is located in the Nayarit-Guerrero, Lower Balsas Basin, and Tehuanan biogeographic districts. Haplotype 13 has the second widest geographical distribution, being found in two Central American biogeographic districts, El Mosquito and Tapachultecan. The rest of the haplotypes are located in a single biogeographic district (Figure 3A,B).

The presence of private haplotypes reflects a high genetic differentiation (*F_ST_* = 0.98) within cpDNA. On the other hand, the differentiation value considering distances between haplotypes (ordered alleles) was *N_ST_* = 0.989 and significantly different from the *G_ST_* value (0.914; *p* < 0.05), which denotes a strong phylogeographic structure. Analysis of molecular variance (Table 1) showed that 99% of the variation is located between populations and 1% of the variation is located within populations. When evaluating the genetic differentiation between the six varieties of *Tillandsia ionantha*, a negative, non-significant value of *F_CT_* was estimated, meaning that there is no variation between the populations of the varieties within the current circumscription of *T. ionantha*. The variation among the eight lineages is the one displaying the highest value of *F_CT_* = 0.98, which reflects the most likely biogeographic scenario. Pairwise *F_ST_* values ranged from 0.0 to 1.0 (Appendix A). In Figure 4A the strong genetic structure between the populations is shown. There are populations that behave as a single genetic population, for example the NIC, GUAS and ZAC populations, are strongly differentiated from the rest of the populations. The Mantel test showed that the genetic differentiation of cpDNA between populations is correlated with geographic distance (*R*2 = 0.272, *p* = 0.0001).

### 2.3. Genetic Diversity and Structure of nDNA

The alignment was 1114 bp long and consisted of 227 nDNA sequences, with 55 polymorphic sites resulting in 88 haplotypes. The estimated haplotype diversity was *Hd* = 0.95 and the nucleotide diversity π = 0.0047. Genetic diversity across all populations (*hT* = 0.972; *vT* = 0.997) was greater than within population average value (*hS* = 0.737; *vS* = 0.244). The 88 haplotypes identified grouped into eight phylogroups strongly associated with biogeographic districts, representing the same pattern observed with cpDNA (Figure 3A,C). The biogeographic districts that present a greater diversity of haplotypes were Oaxaca Plateau, Nayarit-Gerrero + Lower Balsas Basin + Tehuanan, Deciduous Forest of Northern of Veracruz and Tapachultecan + Mosquito. The distribution of these haplotypes is star-shaped; which means, low-frequency haplotypes are derived from higher-frequency haplotypes, separated by a single mutational step.

The presence of private haplotypes reflects a high genetic differentiation (*F_ST_* = 0.75). The differentiation value considering distances between haplotypes (ordered alleles) was *N_ST_* = 0.755 and significantly different from the *G_ST_* value (0.242; *p* < 0.05), which denotes a strong phylogeographic structure. Analysis of molecular variance (Table 1) was similar to that estimated with cpDNA. When analyzing the populations as a single group, it resulted in that 76.59% of the variation is between the populations and only 23.43% of the variation is located within the populations. When evaluating the genetic differentiation between the six varieties of *T. ionantha*, a non-significant low value of *F_CT_* = 0.38 was observed. That is, there is no variation among the populations of the varieties of the current circumscription of *T. ionantha* using the *PHYC* marker. Like the cpDNA, the variation among the eight lineages presented the highest *F_CT_* = 0.78 value. Pairwise *F_ST_* values ranged from 0.0 to 0.936 (Appendix A). However, populations that behave as a single genetic unit and strongly differentiated from the rest of the populations were also observed (Figure 4B). The Mantel test showed that the genetic differentiation of nDNA between populations is correlated with geographic distance (*R*2 = 0.283, *p* = 0.010).

### 2.4. Multispecies Coalescent

The multispecies coalescent analysis supported ten distinct clades (NCclusters = 10): two correspond to the outgroup species and the remaining eight are within the *Tillandsia ionantha* complex and are consistent with the phylogenetic analysis. Figure 5 shows the tree of minimal clusters with high PP values in each of the detected lineages. The gene trees recover as highly supported the eight clades identified with the phylogenetic analyzes (Appendix A). However, support between lineages is moderate.

## 3. Discussion

In the *Tillandsia ionantha* complex, gene trees using three cpDNA regions and a nuclear marker evidenced distinct evolutionary histories in terms of their phylogenetic position within the subgenus but not at the population level, where populations remain reciprocally monophyletic. Several studies have suggested that topological inconsistency between cpDNA and nDNA markers might due to various factors, including convergent evolution, introgression after hybridization, incomplete lineage sorting, and horizontal transfer [23,24,25,26]. For our purposes, and aiming to delimit putative species, both sets of markers identify the same eight lineages with high support values within the *T. ionantha* complex. Three of these eight evolutionary lineages, the *T. ionantha* var. *stricta*, *T. scaposa* and *T. vanhyningii* maintain their morphological, genetic and biogeographic identity. The remaining lineages maintain their genetic and biogeographical identity, but are morphologically cryptic. These results correspond to the allopatric speciation model proposed by Harrison [27], where the process begins with the interruption of gene flow between populations, and then different alleles (or haplotypes) are fixed in the populations, which may or may not acquire different phenotypic characters until they become reciprocally monophyletic. Therefore, these lineages must be considered as evolutionarily independent lineages [8,9].

The diversity and distribution of the haplotypes, the analysis of molecular variation, and the *F_ST_* values for the pairwise comparison among populations with both sets of markers are consistent with the eight clades (lineages) obtained with the phylogenetic analysis. The Mantel test on both sets of markers correlates positively with the geographic distance of the populations, supporting the hypothesis of the allopatric speciation related with the geographical barriers and orographic features of Mexico and Central America that are quite diverse [28,29,30], as these factors lead to a great diversity in climates, vegetation types, and microenvironments [31,32] that allow the geographic isolation of populations and therefore genetic isolation, divergence and the speciation of populations (ecological/allopatric speciation). For *Quercus* [33], *Agave* [34], *Bursera* [35], *Bakerantha* [36], and the *Tillandsia utriculata* complex [37] diversification and divergence of species was suggested to be caused by isolation, expansion, and colonization of new environments.

With our results, we can say that genetic differentiation and population divergence in the *T. ionantha* complex are affected by geographic distance and geographic barriers. Despite the geographical distance, no morphological changes were observed in the reproductive structures [21], which could suggest a niche conservatism in the populations. In contrast to the vegetative traits, species of genus *Tillandsia* have developed a great diversity of floral traits that attract a wide variety of pollinators, including insects, birds, and bats [38,39,40]. However, floral morphology in the *T. ionantha* complex lineages is identical and flowering times overlap; which indicates that they must share the same pollination syndromes and their pollinators must be local and delimited by the same geographical barriers. If so, this hypothesis only supports that geographic barriers prevent the arrival and spread of new individuals/seeds (cpDNA) and pollen (nDNA) from the population of one lineage to the population of another lineage, thus conserving the local gene pool of each lineage. In addition, the high genetic differentiation (*F_ST_*) between the lineages of the *T. ionantha* complex reflect the high rates of gene flow between the populations of the same lineage, thus promoting rapid monophyly at the species level, indicating that gene flow is beneficial for the maintenance of these lineages and their cohesive evolution [41]. In contrast, if geographic barriers were not operative as isolation factors, gene flow would increase between lineages, preventing haplotype fixation and hampering genetic differentiation between detected lineages [41,42].

Multispecies coalescent methods for species delimitation have been highlighted as more objective, robust, and repeatable than traditional morphological taxonomy. However, the use of such methods for the delimitation of species and, later, to make taxonomic decisions is still widely discussed and hesitantly accepted among botanists, especially when cryptic species are discovered [43]. The multispecies coalescent analysis applied in the present work is consistent with the phylogenetic analyses, population structure, and genetic isolation to delimit eight lineages. Three of these eight lineages have morphological characters that, in practical terms, are easy to recognize in the field, while there is no morphological differentiation so far for the remaining five lineages. The cryptic speciation has been increasingly observed in neotropical groups of angiosperm plants [43,44,45], fungi [46,47,48] and animals [49,50,51], suggesting that biodiversity might be underestimated in this region.

Regarding *Tillandsia*, this work is the first to using multilocus data, phylogenetic and phylogeographic analyzes, genetic divergence and differentiation, and the coalescent model to delimit species. It is also perhaps the first work that reports the presence of cryptic species in the genus *Tillandsia*. More work at this same population level will help to identify the presence of more cryptic species in *Tillandsia*. In Bromeliaceae, other studies [52] used multilocus data and phylogenetic and phylogeographic analysis to delimit the two species in the *Tillandsia capillaris* Ruiz and Pav. complex. Romero-Soler et al. [36] used multilocus data, phylogenetic analysis, and the coalescent model to delineate species in the genus *Bakerantha* L.B. Sm. The only two studies that report the presence of other cryptic species in Bromeliaceae are Santos-Leal et al. [53], who through multilocus data, phylogeographic analysis, and coalescent models to identify the presence of a cryptic species in the South American species *Pitcairnia lanuginose* Ruiz and Pav. Goncalves-Oliveira et al. [54] using microsatellite data and phylogeographic analysis reported several cryptic species in *Encholirium spectabile* Mart. ex Schult. f.

Under the general concept of lineage species, which differentiates species as “lineages of metapopulations that evolve separately” [9], in this study we regard the results of the data based on trees, morphological discontinuity [21] and geographic and ecological distribution as providing evidence strong enough to recognize the evolutionary independence between different species of the *Tillandsia ionantha* complex. The current circumscription of the populations and varieties of *T. ionantha* should change according to phylogenetic, genetic isolation and biogeographic evidence here demonstrated. The morphological variation is evident for three lineages within the *T. ionantha* complex. The first is *T. scaposa* that had already been recognized morphologically and its status changed by Ehlers [18] due to the presence of a minute and pedunculated inflorescence, in addition to being ecologically different as it inhabits pine-oak forests in the Guatemalan highlands. *Tillandsia vanhyninigii* differs morphologically from the other varieties due to the caulescent rosette that can measure up to 20 cm in length, the sheath almost the same size as the blade; furthermore, it inhabits rocky walls at the Sumidero Canyon, Chiapas. The populations of the *T. ionantha* var. *stricta* clade are distinctive within the complex by its rosettes, which are compressed and the smallest within the complex, measuring 5–8 (–10) cm [21]; it is restricted in distribution to the Oaxacan Plateau Biogeographic district.

On the other hand there are five lineages that are morphologically indistinguishable, and that we propose to recognize as cryptic species. Three of these lineages had been recognized as varieties of *T. ionantha*. However, the diagnostic characters employed by some authors are variable within and between populations of the different lineages [21]. The diagnostic character of *T. ionantha* var. *maxima* (included in lineage 5) is its large rosettes, which can be twice as large as any other variety of *T. ionantha*. But the size of the rosette is not taxonomically informative in differentiating the evolutionary lineages found in our analysis. The size of the rosettes in other varieties varies between 5 and 12 cm within the same population. If a 5 cm rosette is considered, the double would be 10 cm and this value still falls within the taxonomic varieties of *T. ionantha* and does not exactly diagnose *T. ionantha* var. *maxima*. The diagnostic character in *T. ionantha* var. *zebrina* (included in lineage 1) is the banded pattern of the leaves. This character may or may not be present in individuals within the same population. In addition, we have observed this character in individuals of the *T. ionantha* lineage 2 from Northeastern Mexico, and therefore this character is not taxonomically informative to delimit these lineages. Finally, there are two lineages in the Sinaloan biogeographic district, being morphologically similar to the rest of the cryptic species within this complex.

## 4. Materials and Methods

### 4.1. Sampling Strategy

In order to know the distribution area of the varieties of the *T. ionantha* complex and to select the sampling populations, the type specimens and other specimens deposited in the herbaria CICY, K, F, GH, GOET, M, MEXU, MO, NY, P, SEL, US and WU were revised [55]. The revision of the herbaria allowed us to recognize that the varieties *T. ionanatha* var. *maxima* and *T. ionantha* var. *zebrina* are only known from the type collections (holotypes, isotypes and paratypes). For *T. vanhyningii*. *T. ionantha* var. *stricta*, and *T. scaposa* we found several specimens but they are restricted to biogeographical areas (Table 2); while *T. ionantha* var. *ionantha* has the widest geographical distribution (Figure 6). Nineteen populations, each with 10 to 21 individuals were selected (Table 2), leaves were collected and stored in silica gel. The selected populations included the type localities for each variety of *T. ionantha*, *T. vanhyningii* and *T. scaposa*. Similarly, we selected populations to cover the whole geographical distribution of *T. ionantha* var. *ionantha* (Figure 6).

### 4.2. DNA Isolation, Amplification, and Sequencing

DNeasy Plant Mini Kit (QIAGEN) was used to extract DNA from 240 individuals from the 19 populations (Table 2) and 35 outgroup species [22], The sequences will be deposited in the Genbank (if you want these sequences contact the corresponding author). The out-group species selection was based on previous phylogenetic studies of Barfuss et al. [56], and Pinzón et al. [37].

Three chloroplast (cpDNA) spacers (*rps16-trnQ*, *rpl16-rps3* and *trnT-trnL-trnF*) and one low copy nuclear (nDNA) gene (*PHYC*) were amplified. The selection of markers were based on Pinzón et al. [37], Barfuss et al. [56], Castello et al. [52], Barfuss [57], and Rodríguez-Figueroa [22]. These markers were selected for being the most informative for *Tillandsia*. Appendix A specifies the names and sequences of the primers used in the amplification and the PCR conditions. The *rps16-trnQ*, *rpl16*-*rps3* and *trnT-trnL-trnF* markers were amplified in a single fragment and various internal primers were used for sequencing (Appendix A). The Polymerase Chain Reaction (PCR) was performed in a final volume of 20 µL. The mix consisted of 10 µL of Phusion Hot Start II High-Fidelity PCR Master Mix (Thermo Fisher Scientific, Waltham, MA, USA); 1 µL of each primer (forward and reverse) at a concentration of 10 µM; 2 µL of 0.04% BSA; 5 µL ultrapure water; and 1 µL of DNA (2–8 µg/µL). The PHYC molecular marker was amplified and sequenced into two overlapping fragments (Appendix A). For PHYC the PCR was performed in a final volume of 10 µL. The mix consisted of 5 µL de GoTaq^®^ Green Master Mix (Promega Corporation, Madison, WI, USA); 1 µL of each primer (forward and reverse) at a concentration of 10 µM; 1 µL of 0.04% BSA; 2.2 µL ultrapure water; and 1 µL de DNA (2–8 µg/µL). The PCR products were purified and sequenced at Macrogen, South Korea. The sequences were assembled using Sequencher, aligned with ClustalW [58] implemented in MegaX [59] and manually edited in BioEdit [60].

### 4.3. Phylogenetic Analyses

To explore the variability of each marker, we calculated the number of variable and potentially parsimony informative sites using MEGA11: Molecular Evolutionary Genetics Analysis version 11 [59]. Two phylogenetic reconstructions analyses were performed using MrBayes 3.2.7 software [61], one with the cpDNA matrix data and the other with the nDNA matrix data. For each matrix data we used the estimated nucleotide substitution model in MegaX (Appendix A). We performed two independent runs with 10,000,000 generations for each matrix data, sampling every 1000 generations. Each run consisted of one cold and three hot chains, unbinding the “tratio”, “statefrq” and “shape” parameters on all partitions, as well as setting “rate” as a variable for all partitions. A 50% majority rule consensus tree was constructed after discarding the initial 25% of trees. For the Maximum Likelihood approach, we ran 10 replications to retrieve the best topology in RAxML 8.2.12 [62] and a 1000 replications Bootstrap for nodal support using the CIPRES Science Gateway 3.3 [63], with all other parameters as default. The trees obtained were visualized and edited using FigTree [64].

### 4.4. Genetic Diversity and Structure Analyzes

Like the phylogenetic analyses, we performed the genetic structure analyses separately for both sets of markers. The number of haplotypes (*H*), the diversity of haplotypes (*Hd*), the diversity of nucleotides (*Π*) and polymorphic sites were estimated using the DNSpV.6 program [65]. As each of the eight evolutionary lineages found in the phylogenetic analyzes are restricted to specific biogeographical districts. In this section, we use PopART version 1.7 [66] and the TCS network method [67] to analyze the evolutionary relationships between haplotypes and biogeographic districts where each of the studied populations are distributed (Table). Haplotype distribution patterns were plotted on a map. The indices for ordered (*vS*, *vT*) and unordered (*hS*, *hT*) haplotypes and differentiation parameters *N_ST_*, *G_ST_* [67] and *N_ST_* [68,69] were estimated using PERMUT v.1.0 [69]. The significant difference between the *N_ST_* and *G_ST_* values was tested with 10,000 permutations. A significant value of *N_ST_* indicates phylogeographic structure and that the most similar haplotypes are geographically close to each other.

We evaluated the population structure with pairwise *F_ST_* values calculated with ARLEQUIN version 3.5 [70] and PAST version 4.08 [71] was used to represent the values in a matrix plot. Analysis of molecular variance (AMOVA) [72] were also performed using ARLEQUIN v.3.5 to test the structure between populations using both data sets separately and different hierarchical levels: eight lineages detected by phylogenetic analysis and the current taxonomic classification of the *T. ionantha* complex [17]. To determine whether divergence among populations is an effect of isolation by distance, we correlated pairwise genetic distance (measured as *F_ST_*) versus geographical distance matrices by a Mantel test using the Software GenAlEx 6.5 [73].

### 4.5. Species Delimitation Using STACEY

A Bayesian coalescent method of species delimitation was performed using STACEY (species tree estimation using DNA sequences from multiple loci) [74], which is an extension of DISSECT [75]. STACEY was implemented in BEAST ver. 2.4.4 [76,77,78].

We ran STACEY using a multilocus dataset (ptDNA and nDNA) with all populations of the *T. ionantha* complex and two outgroup species. Input xml files were prepared in BEAUti v. 2.4.7 [76] and the corresponding substitution models were set according to results from MegaX (Appendix A). The analysis were run for 100 million generations of the MCMC chains, sampling every 5000 generations. Convergence of the stationary distribution was checked by visual inspection of plotted posterior estimates using Tracer ver. 1.7 [79]. After discarding the first 1000 trees as burn-in, the samples were summarised in the maximumclade credibility tree using TreeAnnotator ver. 1.6.1 [77] with a posterior probability limit of 0.5 and summarising of mean node heights. The results were visualised using FigTree ver. 1.3.1 [64].

## 5. Conclusions

Multilocus data and different analysis methods support species delimitation and cryptic diversity in *T. ionantha*. Morphological and genetic evidence corroborates the hypothesis of the evolutionary lineage of *Tillandsia scaposa*, *T. vanhyningii* and *T. ramireziana* (=*T. ionantha* var. *stricta*). On the other hand, the remaining five lineages are supported by genetic evidence, but not by morphology. Even though these five cryptic species do not have unambiguous macromorphological characters, the hypothesis that they are populations that have speciated is supported by the occurrence of clear phylogenetic structure (high levels of divergence between the long branches of the phylogenetic trees and high values of support in nodes), geographical and ecological differences. However, for practical reasons and identification of the populations in the field, we will avoid describing and naming them until we have more morphological evidence. Further studies can explore micromorphological and anatomical characters to corroborate these hypotheses of cryptic lineages in *T. ionantha*.

## 6. Taxonomy

*Tillandsia scaposa* is validated as an evolutionary lineage as proposed by Ehlers [17]; and two varieties of *T. ionantha* are treated at the species level: *T. vanhyningii* and *T. ramireziana*. Likewise, it is proposed that these taxa be eliminated from the *T. ionantha* complex since the morphological and molecular evidence support this. In such a way that the complex is currently composed of five morphologically cryptic species.

***Tillandsia scaposa*** (L.B.Sm.) Ehlers in Bromelie 1999 (3): 69.1999

≡ *Tillandsia ionantha* var. *scaposa* L.B.Sm. in Lilloa 6: 384, t. 1, Figure 7. 1941—Lectotype (designated by Smith and Downs, 1977): Guatemala, Sacatepéquez, near Antigua, alt. 1500–1600 m, Nov 1938–Feb 1939, Standley 63065 (GH!; isolectotype: F!).

***Tillandsia ramireziana*** J.J. Ancona and Pinzón *nom.* and *stat. nov*.

Basionym: *T. ionantha* var. *stricta* Koide f. *stricta* Koide in J. Bromeliad Soc. 43: 162. 1993.―Holotype: Mexico: Oaxaca; 6 km east of El Camaron, elev. ca. 2000 m. Epiphytic on Quercus sp. Koide and Schuster sn. legit., April 1982. Flowered in cultivation, P Koide s.n. (SEL!; isotypes: US!, MEXU!).

=*T. ionantha* var. *stricta* Koide f. *fastigiata* Koide in J. Bromeliad Soc. 43: 162. 1993.—Holotype: Mexico: Oaxaca; 6 km east of El Camaron, elev. ca. 2000 m. Epiphytic on Quercus sp., growing intermingled with typical 71 ionantha var. stricta, Koide and Schuster s.n. legit., April 1982. flowered in cultivation, P Koide sn. (SEL!; US!, MEXU!).

Note. *Tillandsia ramireziana* is a replacement name for *T. stricta*, since the name already exists to name for species that is distributed in Brazil (*Tillandsia stricta* Sol. ex Sims). This name is restricted to the populations that are distributed in the Oaxacan Plateau and the Chiapas Highlands (Figure 7). The vegetation is a low deciduous forest type in transition with oak forests in elevation from 200 to 1600 m. Within this species, Koide [80] described two morphological forms. However, our analysis does not recognize the delimitation of both forms as lineages; therefore both names are recognized as synonyms of *T. ramireziana*.

**Eponymy**. The specific epithet of this species is in honor to Dra. Ivón Ramírez Morillo for her long career in the taxonomic and systematic studies on Bromeliaceae.

***Tillandsia vanhyningii*** (B.T. Foster) Beutelspacher and García-Martínez

≡ *T. ionantha* var. *vanhyningii* B.T. Foster in Bull. Bromeliad Soc. 7: 71, f. 1957―Holotype: Mexico: Chiapas: vert ledges overhanging Rio Grijalva, 6 Apr 1957, Foster and Van Hyning 2957 (US!).

## Figures and Tables

**Figure 1 plants-11-01706-f001:**
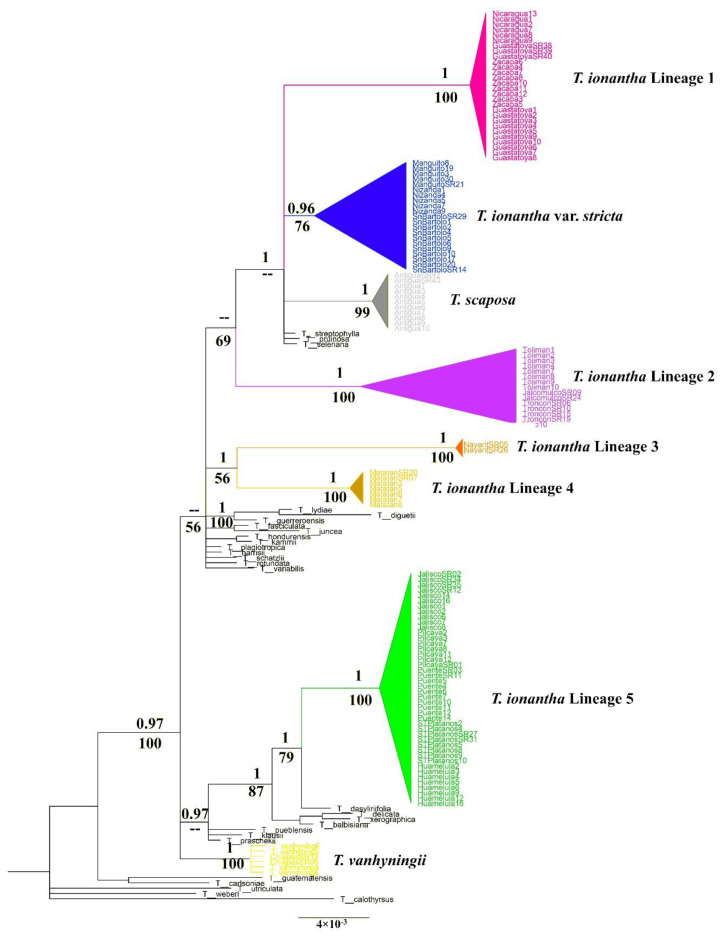
Phylogenetic tree estimated with Bayesian Inference using three cpDNA markers; PPB values ≥ 0.95 on the branches; Bootstrap values ≥ 50 below the branches. The collapsed and colored clades represent each of the eight lineages within the *T. ionantha* complex. Uncolored branch terminals (taxa) represent outgroup species.

**Figure 2 plants-11-01706-f002:**
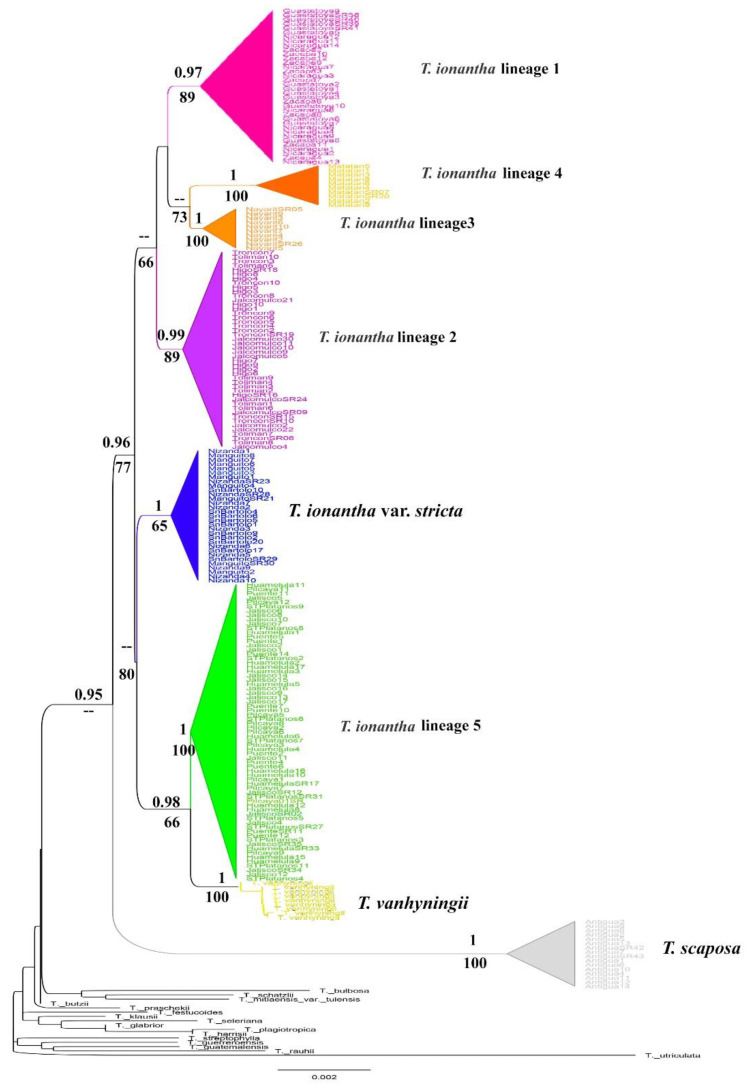
Phylogenetic tree estimated with Bayesian Inference using PHYC; PPB values ≥ 0.95 on the branches; Bootstrap values ≥ 50 below the branches. The collapsed and colored clades represent each of the eight lineages within the *T. ionantha* complex. Uncolored branch terminals represent outgroup species.

**Figure 3 plants-11-01706-f003:**
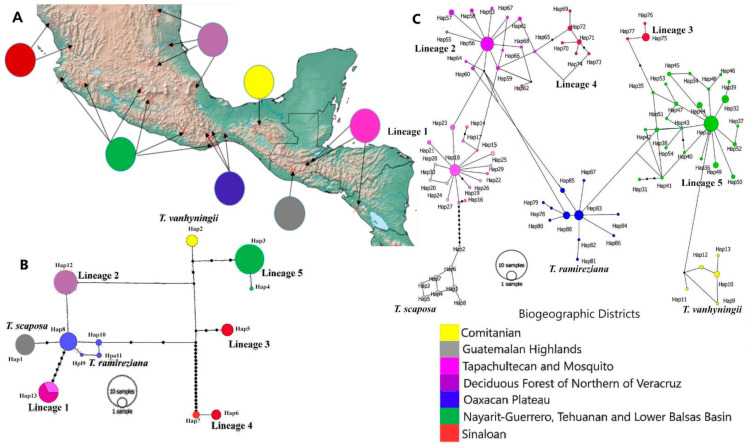
Evolutionary relationships between haplotypes, each set of haplotypes of the same color represents the same lineage. (**A**) Geographic distribution pattern of the haplotypes for both sets of markers. (**B**) Reconstruction of the haplotype network of the markers rps16-trnQ and rpl16-rps3 using the TCS Network model. (**C**) Reconstruction of the PHYC marker haplotype network using the TCS network model.

**Figure 4 plants-11-01706-f004:**
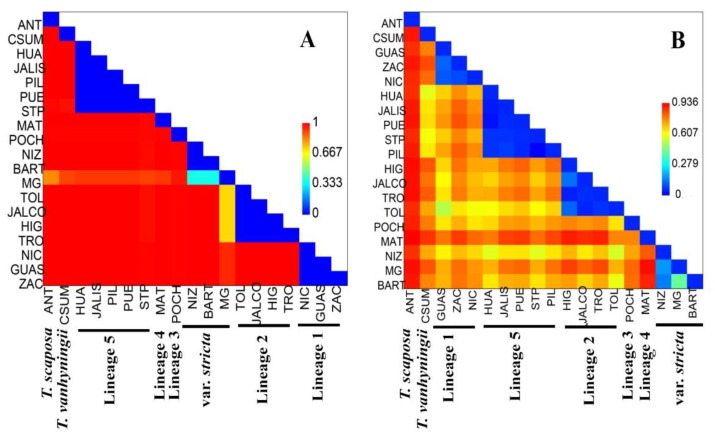
*F_ST_* values for pairwise comparison between populations of *T. ionantha* complex. (**A**) *F_ST_* pairwise of cpDNA and (**B**) *F_ST_* pairwise of nDNA.

**Figure 5 plants-11-01706-f005:**
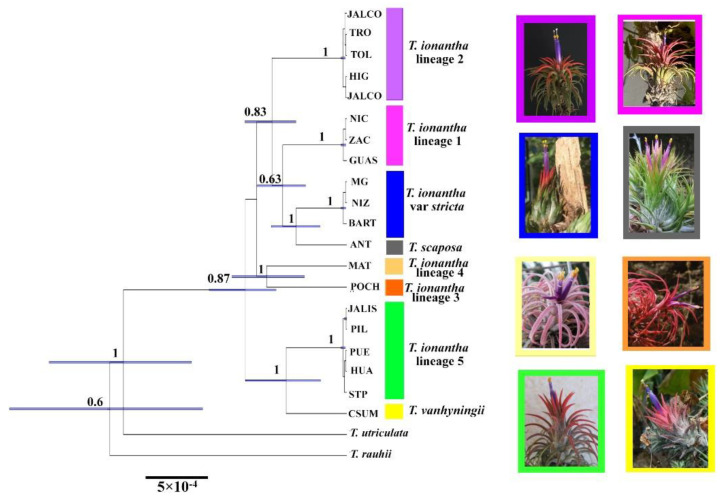
Maximum clade credibility tree of the *Tillandsia ionantha* species aggregate, along with outgroup taxa, resulting from the multilocus coalescent analyses in STACEY (Beast2). Values above branches are posterior probabilities of clade support. The colors of the perimeter of the photographs correspond to the colors of each of the evolutionary lineages of the minimal clusters tree.

**Figure 6 plants-11-01706-f006:**
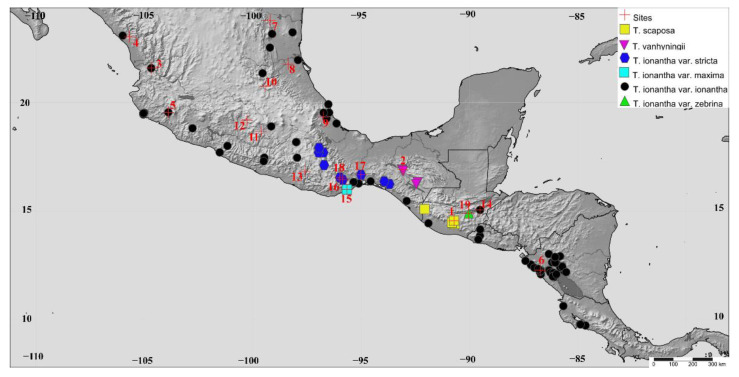
Geographic distribution of the taxa of the *T. ionantha* complex and populations sampled (numbers 1–19). Populations 1, 2, 16, 18 and 19 represent the type localities of *T. scaposa*, *T. vanhyningii*, *T. ionantha* var. *zebrina*, *T. ionantha* var*. stricta and T. ionantha* var*. maxima* respectively.

**Figure 7 plants-11-01706-f007:**
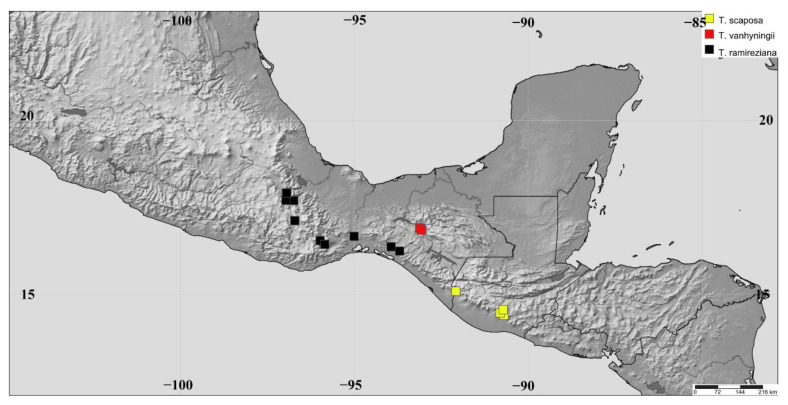
Geographical distribution of *Tillandsia ramireziana* (black squares), *Tillandsia scaposa* (yellow squares) and *Tillandsia vanhyningii* (red squares).

**Table 1 plants-11-01706-t001:** Analysis of molecular variance (AMOVA) based on cpDNA and nDNA sequences. * = five varieties of *T. ionantha* + *T. scaposa*.

Model	Source Variation	D.F.	Variation (%)	*F*-Statistic	*p* Value
	*cpDNA*				
All Populations	Among populations	18	99.08	*F_ST_* = 0.99	1 × 10^−5^
	Within populations	210	0.92		
Six Taxa *	Among groups	5	−9.46	*F_CT_* = −0.09	0.44
	Among populations within groups	13	100.58	*F_ST_* = 0.99	0.0001
	Within groups	210	0.96	*F_SC_* = 0.99	1 × 10^−5^
Eigth Lineages	Among groups	7	98.89	*F_CT_* = 0.98	1 × 10^−5^
	Among populations within groups	11	0.31	*F_ST_* = 0.99	1 × 10^−6^
	Within groups	210	0.8	*F_SC_* = 0.28	1 × 10^−6^
	*nDNA*				
All Populations	Among populations	18	76.59	*F_ST_* = 0.765	0.0001
	Within populations	208	23.42		
Six Taxa *	Among groups	5	38.94	*F_CT_* = 0.38	0.13
	Among populations within groups	13	42.2	*F_ST_* = 0.81	1 × 10^−5^
	Within groups	208	18.887	*F_SC_* = 0.69	1 × 10^−5^
Eigth Lineages	Among groups	7	78	*F_CT_* = 0.78	1 × 10^−6^
	Among populations within groups	11	1.15	*F_ST_* = 0.79	1 × 10^−5^
	Within groups	208	20.86	*F_SC_* = 0.052	1 × 10^−6^

**Table 2 plants-11-01706-t002:** Taxa, populations, code and biogeographical districts of the individuals collected. * = type locality. Ele. = Elevation. Biogeographical districts *sensu* Morrone [28,29]. N = Individuals collected.

Taxa	Population	Code	Biogeographical District	Geographic Location	Ele.	N
*T. scaposa*	1. Antigua *	ANT	Guatemala Highlands	14°33′17″ N, 90°43′14″ W	1740 m	16
*T. vanhyningii*	2. Cañón del Sumidero *	CSUM	Comitanian	16°54′39″ N, 93°05′40″ W	1100 m	11
*T. ionantha* var. *ionantha*	3. Pochochitán	POCH	Sinaloan	21°34′52″ N, 104°40′07” W	700 m	12
	4. Matatán	MAT	Sinaloan	23°1′42″ N, 105°40′45″ W	131 m	12
	5. Jalisco	JALIS	Nayarit-Gerrero	19°48′44″ N, 105°16′27″ W	49 m	21
	6. Nicaragua	NIC	Tapachultecan	12°16′17″ N, 86°44′07″ W	24 m	14
	7. El Troncón	TRO	Deciduous Forest of Northern of Veracruz	23º46′42″ N, 99º12′22″ W	700 m	13
	8. El Higo	HIG	Deciduous Forest of Northern of Veracruz	21°45′57″ N, 98°22′11″ W	48 m	12
	9. Jalcomulco *	JALCO	Deciduous Forest of Northern of Veracruz	19°20′06″ N, 96°45′10″ W	355 m	12
	10. Barranca de Tolimán	TOL	Deciduous Forest of Northern of Veracruz	20°44′38″ N, 99°26′09″ W	1620 m	10
	11. Pilcaya	PIL	Lower Balsas Basin	18°40′01″ N, 99°36′35″ W	1300 m	11
	12. Santo Tomás de los Plátanos	STP	Lower Balsas Basin	19°10′59″ N, 100°15′37″ W	1562 m	12
	13.- El Puente	PUE	Lower Balsas Basin	16°49′19″ N, 97°35′11″ W	1250 m	12
	14. Zacapa	ZAC	Mosquito	15°02′58″ N, 89°31′14″ W	200 m	10
*T. ionantha* var. *maxima*	15. San Pedro Huamelula *	HUA	Tehuanan	15°59′40″ N, 95°39′54″ W	70 m	15
*T. ionantha* var. *stricta*	16. El Manguito *	MG	Oaxaca Plateau	16°32′35″ N, 95°59′03″ W	1120 m	11
	17. Nizanda	NIZ	Oaxaca Plateau	16°39′58″ N, 95°00′28″ W	180 m	11
	18. San Bartolo	BART	Oaxaca Plateau	15º26′05″ N, 95º50′58″ W	1120 m	11
*T. ionantha* var. *zebrina*	19. Guastatoya *	GUAS	Mosquito	14°53′16″ N, 90°02′23″ W	460 m	14

## Data Availability

Not applicable.

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
