# Peer review of "Multilocus Data Analysis Reveal the Diversity of Cryptic Species in the Tillandsia ionantha (Bromeliaceae: Tillansiodeae) Complex"

_plants, 2022, doi:10.3390/plants11131706_

Round 1
Reviewer 1 Report
Ancona and colleagues report a nice study aiming at delimiting independent evolutionary lineages (species) in the Tillandsia ionantha species complex. To archive the goals, the authors use sequence data from three chloroplast markers and one nuclear region on a satisfactory number of samples and populations, covering the whole distribution range and morphological variation of the T. ionantha species complex. They apply phylogenetic approaches (including coalescent-based species tree inference) along with methods aiming at evaluating the between- and within- populations genetic structure and the study of the molecular variance. Results from phylogenetic (gene-tree) methods and those from genetic structure analyses were used to delimit “independent evolutionary lineages”. These were subsequently used to assign samples to species and estimate the species tree in *BEAST.
The manuscript is submitted for publication in the special issue of Plants entitled “integrative taxonomy of plants”. I found the study being a nice contribution for the special issue. However, I will suggest publication after a deep and careful revision.
Main issues:
1) The study does not really use integrative approaches, since it makes us of molecular data (DNA sequences) as only source of evidence. I don’t think this is a problem, since other papers I have seen accepted for publication in the special issue do the same. However, I would at least delete the word “integrative” from the title.
2) In my opinion, the main issue is the use of coalescent-based species delimitation methods. This is stated more than once throughout the manuscript, although no true species-delimitation methods are used. The authors used *BEAST, which is a coalescent-based method, but not for species delimitation. *BEAST infers the species tree based on multilocus data, and the assignment of samples to species (species delimitation) needs to be provided a-priori by the users. The authors inferred species delimitation based on the results of the phylogenetic trees (the gene trees…MrBayes and RAxML analyses) and of those from the “population structure” analyses and use it (the inferred species delimitation) in *BEAST. However, this does not mean to perform a coalescent-based species delimitation analyses.
The paper is submitted to a special issue on “integrative taxonomy of plants” and I think it would really benefit from a real species delimitation analysis. There are methods capable of doing that in a bayesian framework. STACEY - for example - is a species delimitation method also implemented in BEAST. The authors could have used it instead of performing two analyses of *BEAST with different assignments of samples to species.
Since both methods are implemented in the software BEAST, I don't think it will coast much effort or extra-work to make those analyses (just adapting the input and run the analyses). A STACEY (or BPP) analysis would really make the results sounder. If the authors don’t think these analyses are necessary (and if the editor think that it is fine so), then they should at least avoid using the term “coalescent-based species delimitation” (and similar) throughout the text.
3) The manuscript would really benefit from a careful revision of the writing form. It is not only about the English form, it is also about several type errors that could have been avoided by carefully reading and polishing the manuscript. For example, at lines 527-530 the same sentence is written twice. At line 667 the sentence finish with “Tillandsia_2022”, which does not seem to be related to anything. Sometime Figures are cited wrongly (e.g., at line 351, Figure 5B should be Figure 4B). The term “coalescent models” (plural) is repeated throughout the manuscript (it should be singular).
Please check carefully and improve the writing form. I have uploaded a file with some suggestions (some…still a lot more could be improved).
4) Figure 1. This tree (and the one in Figure 2) are represented with the clades of the “main lineages” collapsed and coloured. I think it would be recommendable to place the original trees (showing also resolution within these clades) as Supporting Materials (Figures SXXX…). Caption needs to be improved and extended. Please, indicate the meaning of the colours of the collapsed clades (trivial but necessary). Accessions that are not coloured are not being part of the T. ionantha complex? Such information need to be provided.
5) Figure 2. As above, please improve the caption. Moreover, what do the different blue tones of some accessions of T. ionantha var. stricta mean?
And also, is the clade (lineage1, (lineage 4, lineage3)) supported? If not, then please put "--" above and below the corresponding branch, as it is done for other unsupported clades.
6) Figure 3. I find colours of this figure a bit confusing. Haplotypes should be better coloured according to haplotype-group/lineage, not according to biogeographic district. This lead to some shortcomings, e.g.:
-It would be nice to see in the map which population belong to lineage 3 and 4, but this is not possible since they have the same colour.
- Also, some of the populations have haplotypes from lineage 1 and lineage 2, is it possible?. By the way, it seems there are more tones of purple in the map than in the legend or in the haplotype networks. How is it possible?
- Are circles in the map meant to be populations?
- Populations of T. scaposa are missing from the map.
This is a very important figure of the paper, and I think it can be improved by solving these small issues or (if I misunderstood) providing more information in the caption.
Material and Methods.
Lines 558-561: It would be good to have listed somewhere samples and sequences (e.g., GenBank accession numbers) that has been used as outgroup; rather as table (maybe easier) or in the text. For example, how many outgroup species have been included as outgroup? How many accessions per species? This will make the interpretation of Figure 1 and 2 much easier.
Line 662: It is stated that the *BEAST analyses were run for 1007 generations. This is a very big number and I guess the authors meant 100,000,000 generations, which is 10*107 (=108, not 1007). Please, correct it, otherwise clarify!
Data Availability: It is good practise to make (at least) the DNA sequences available in a public repository, such us the GenBank, the European nucleotide archive, or similar.
This is true for the sequences produced for this study. For the ones already used in other studies, it would be nice to have a list with GenBank accession numbers (see also the above comment).
Figures S1-S5. Please, state in the caption that values above branches are posterior probabilities. Please write names of marker regions in italics.
Figure S5. “Maximum credibility tree” instead on “tree of maximum credibility”. Again, state in the caption that values above branches are posterior probabilities.
Table S3. References mentioned in the caption are not listed in the reference list or anywhere else.
First line, 5th column. Maybe better to write directly “length” instead of “long”.
In the same column, please express lengths in bp (base pairs), not in pb.
For the first two regions, the parsimony informative sites are more than the variable sites. Shouldn’t they be a fraction of the variable sites? Please verify!
For several additional small issues, I am providing an annotated pdf file of the manuscript

Author Response
Reviewer 1
1.- The study does not really use integrative approaches, since it makes us of molecular data (DNA sequences) as only source of evidence. I don’t think this is a problem, since other papers I have seen accepted for publication in the special issue do the same. However, I would at least delete the word “integrative” from the title.
R= We agree and have removed the word "integrative" from the title.
2.- In my opinion, the main issue is the use of coalescent-based species delimitation methods. There are methods capable of doing that in a bayesian framework. STACEY - for example - is a species delimitation method also implemented in BEAST. The authors could have used it instead of performing two analyses of *BEAST with different assignments of samples to species.
R= Coalescing species delimitation analysis was performed using STACEY for BEAST.
3.- The manuscript would really benefit from a careful revision of the writing form. It is not only about the English form, it is also about several type errors that could have been avoided by carefully reading and polishing the manuscript. For example, at lines 527-530 the same sentence is written twice. At line 667 the sentence finish with “Tillandsia_2022”, which does not seem to be related to anything. Sometime Figures are cited wrongly (e.g., at line 351, Figure 5B should be Figure 4B). The term “coalescent models” (plural) is repeated throughout the manuscript (it should be singular).
R= We have paid more attention to the grammatical form and have corrected the reviewer's suggestion
4.- Figure 1. This tree (and the one in Figure 2) are represented with the clades of the “main lineages” collapsed and coloured. I think it would be recommendable to place the original trees (showing also resolution within these clades) as Supporting Materials (Figures SXXX…). Caption needs to be improved and extended. Please, indicate the meaning of the colours of the collapsed clades (trivial but necessary). Accessions that are not coloured are not being part of the T. ionantha complex? Such information need to be provided.
R= We believe that presenting the original (uncollapsed) trees does not help the readers much, because there is no resolution within the lineages, no groups with high support values that reflect evolutionary patterns are observed.
We expanded the description of the captions of figure captions. Explaining the colors and taxa in black letters: The collapsed and colored clades represent each of the eight lineages within the T. ionantha complex. Uncolored branch terminals represent outgroup species.
5.- Figure 2. As above, please improve the caption. Moreover, what do the different blue tones of some accessions of T. ionantha var. stricta mean?
And also, is the clade (lineage1, (lineage 4, lineage3)) supported? If not, then please put "--" above and below the corresponding branch, as it is done for other unsupported clades.
R= The shades of the blue clade were unified, and the values of the PP and Bootstrap of the clade (lineage1, (lineage 4, lineage3)) were placed.
6.- Figure 3. I find colours of this figure a bit confusing. Haplotypes should be better coloured according to haplotype-group/lineage, not according to biogeographic district. This lead to some shortcomings, e.g.:
-It would be nice to see in the map which population belong to lineage 3 and 4, but this is not possible since they have the same colour.
- Also, some of the populations have haplotypes from lineage 1 and lineage 2, is it possible?. By the way, it seems there are more tones of purple in the map than in the legend or in the haplotype networks. How is it possible?
- Are circles in the map meant to be populations?
- Populations of T. scaposa are missing from the map.
R= We modify figure 3. Figure 3. Evolutionary relationships between haplotypes, each set of hoplatypes of the same color represents the same lineage. (A) Geographic distribution pattern of the haplotypes for both sets of markers. (B) Reconstruction of the haplotype network of the markers rps16-trnQ and rpl16-rps3 using the TCS Network model. (C) Reconstruction of the PHYC marker haplotype network using the TCS network model.
Material and Methods.
7.- Lines 558-561: It would be good to have listed somewhere samples and sequences (e.g., GenBank accession numbers) that has been used as outgroup; rather as table (maybe easier) or in the text. For example, how many outgroup species have been included as outgroup? How many accessions per species? This will make the interpretation of Figure 1 and 2 much easier.
R= We are pending uploading the sequences to GenBank.
8.- Line 662: It is stated that the *BEAST analyses were run for 1007 generations. This is a very big number and I guess the authors meant 100.000.000 generations, which is 10*107 (=108, not 1007). Please, correct it, otherwise clarify!
R= We change 1007 for 100.000.000
9.- Figures S1-S5. Please, state in the caption that values above branches are posterior probabilities. Please write names of marker regions in italics.
R= We make the suggested changes.
10.- Figure S5. “Maximum credibility tree” instead on “tree of maximum credibility”. Again, state in the caption that values above branches are posterior probabilities.
R= We make the suggested changes.
11.- Table S3. References mentioned in the caption are not listed in the reference list or anywhere else.
R= References are cited below the supplement table and in the References of the manuscript.
12.- First line, 5th column. Maybe better to write directly “length” instead of “long”. In the same column, please express lengths in bp (base pairs), not in pb.
R= we made the suggested corrections.
13.- For the first two regions, the parsimony informative sites are more than the variable sites. Shouldn’t they be a fraction of the variable sites? Please verify!
R= we made the suggested corrections

Reviewer 2 Report
In the manuscript "Mulitlocus data and integrative analysis reveal the diversity of cryptic species in the Tillandsia ionantha (Bromeliaceae: Tillansiodeae) complex," authors Ancona et al. investigated species diversity, including what potentially looked like cryptic diversity based on a previous study. The authors apply Sanger sequencing phylogenetic and population approaches to determine that the T. ionantha complex consists of eight lineages, five of which are cryptic.
The authors apply Sanger sequencing of three cpDNA markers and one nuclear gene. Although the field has largely moved towards high throughput sequencing, I agree with the authors that there is adequate molecular variation to answer their questions. The authors provided strong evidence that the lineages were distinct with multiple data sources and analyses.
I provide a critical review of the manuscript below, starting with larger conceptual issues and then ending with minor details.
The cpDNA and nDNA overlap in some of the patterns, for example, reconstructing eight lineages, but are inconsistent in the relationships among those lineages and close relatives. I feel that this needs to be discussed more. One explanation for what is happening might be that the haploid cpDNA coalesces faster than the diploid nDNA. But the results leave much to be interpreted. For example, the cpDNA suggests the best taxonomic strategy would be to rename all of the lineages as distinct species, but the nDNA suggests that the best strategy might be to name them all T. ionantha or as separate species.
How were the cpDNA markers treated in the starBeast analysis? Were they treated as independent markers (which they shouldn't be since they are inherited as a single unit) or combined? Furthermore, how were the different ploidy levels of the two genomic data dealt with in the starBeast analyses? The ploidy levels are quite important to consider in coalescence models.
The second model defined on lines 656-658 is not clear to me. What previous result are the authors referring to? Furthermore, why isn't a monophyletic T. ionantha being tested? I worry that testing only two models is too simplistic and if we are asking the model to differentiate between two suboptimal models, we will not be recovering the optimal model and overconfidently accepting the suboptimal model.
I was a bit dubious about the demarcation of populations into ten biogeographic districts on line 630. This is either a subjective or posthoc decision, either of which is problematic. I would like to see an objective a priori criterion applied. For example, why not use a program like Structure that infers population structure? Such an approach would lend objectivity and is incredibly important because if the populations are subjectively clustered and don't represent biologically meaningful groups, measures like Fst are meaningless.
In the introduction on lines 95-96, the research question "is cryptic species a common phenomenon" is asked. I don't think this is an appropriate question and the authors certainly do not test whether it is "common." A simple solution would be to rewrite the question to something like "Are cryptic species present in the T. ionantha complex?", which is more in line with what the study answers without trying to define what is meant by "common."
The naming convention is not consistent in the manuscript. On line 130, it is stated that the T. ionantha is monophyletic according to the PHYC marker, but when looking at Figure 2, it is not monophyletic, rather paraphyletic, because the authors are treating T. vanhyniningii at the specific level. This is a small issue, but needs to be rectified.
I did not find the Abstract set up well. Lines 18-20 do not set the study up in a logical way and rewriting should be considered.
How was model selection conducted? Was incongruence assessed and if so, how? For all of the Bayesian analyses (MrBayes and Beast), there is no mention of how both stationarity and convergence were assessed.
I realize that English might not be the first language of some of the authors and made corrections for some of the grammatical issues below. Overall I thought the manuscript was well written.
Minor issues:
Line 19: Add "they" before "serve"
Line 24: Add "analyses" after "distance"
Line 26: Change "by" to "of"
Line 55: A subject is missing from this sentence after the word "allow," perhaps include "us", "botanists", or "biologists"
Lines 66-67: The rosette is said to be less than 20 cm first, but then up to 20 cm on the next line. It cannot be both less than 20 cm and 20 cm.
Line 94: Change "between" to "among"
Line 317: The sentence reads awkwardly. I would say "The alignment was 1114 bp long and consisted of 227 nDNA sequences" or something like that.
Line 376: The grammar needs to be fixed in this sentence.
Figure 5: The image looks stretched vertically.
Line 450: Add "model" after "allopatric"
Line 461: Remove the word "Well,"
Line 462: I would be careful about invoking "niche conservatism" here as it really isn't tested.
Lines 491-493: Didn't Granados Mendoza et al. 2017 Botany 95(7) do this in a group of Tillandsia? Note that I am not an author on that paper.
Line 526: Remove the word "very"
Figure 6: I cannot see the numbers
Table 2: This table is messy, please fix the alignments.
Line 613: Fix the spelling of MrBayes
Line 622: Add "set to" after "parameters"
Line 667: Remove Tillandsia_2022
Author Response
Reviewer 2
The cpDNA and nDNA overlap in some of the patterns, for example, reconstructing eight lineages, but are inconsistent in the relationships among those lineages and close relatives. I feel that this needs to be discussed more. One explanation for what is happening might be that the haploid cpDNA coalesces faster than the diploid nDNA. But the results leave much to be interpreted. For example, the cpDNA suggests the best taxonomic strategy would be to rename all of the lineages as distinct species, but the nDNA suggests that the best strategy might be to name them all T. ionantha or as separate species.
R= In this work, the inconsistencies between both markers are not discussed in depth because our objective is to delimit species. Dr. Juan Pablo Pinzón, co-author of this work and director of the Project "Systematics and phylogeography of the Tillandsia ionantha complex (Bromeliaceae) under the paradigm of the integrative taxonomy" is currently preparing a manuscript where he studies the phylogenetic position of the T. ionantha in the subgenus Tillandsia. In the work of Dr. Pinzón, this problem of inconsistency and its phylogenetic relationships with other similar species are discussed in greater depth.
How were the cpDNA markers treated in the starBeast analysis? Were they treated as independent markers (which they shouldn't be since they are inherited as a single unit) or combined? Furthermore, how were the different ploidy levels of the two genomic data dealt with in the starBeast analyses? The ploidy levels are quite important to consider in coalescence models.
R= They were treated as independent, although they are inherited in a single unit, the nucleotide substitution patterns are different for each cpDNA marker. For this reason, we decided to treat them as different units.
I was a bit dubious about the demarcation of populations into ten biogeographic districts on line 630. This is either a subjective or posthoc decision, either of which is problematic. I would like to see an objective a priori criterion applied. For example, why not use a program like Structure that infers population structure? Such an approach would lend objectivity and is incredibly important because if the populations are subjectively clustered and don't represent biologically meaningful groups, measures like Fst are meaningless.
R= As each of the eight evolutionary lineages found in the phylogenetic analyzes are restricted to specific biogeographical districts. In this section, we use popart version 1.7 [66] and the tcs network method [67] to analyze the evolutionary relationships between haplotypes and biogeographic districts where each of the studied populations are distributed. Haplotype distribution patterns were plotted on a map.
In the introduction on lines 95-96, the research question "is cryptic species a common phenomenon" is asked. I don't think this is an appropriate question and the authors certainly do not test whether it is "common." A simple solution would be to rewrite the question to something like "Are cryptic species present in the T. ionantha complex?", which is more in line with what the study answers without trying to define what is meant by "common."
R= We changed the formulation of the question.
The naming convention is not consistent in the manuscript. On line 130, it is stated that the T. ionantha is monophyletic according to the PHYC marker, but when looking at Figure 2, it is not monophyletic, rather paraphyletic, because the authors are treating T. vanhyniningii at the specific level. This is a small issue, but needs to be rectified.
R= monophyletic was changed to paraphyletic.
I did not find the Abstract set up well. Lines 18-20 do not set the study up in a logical way and rewriting should be considered.
R= The first lines of the abstract were rewritten.
How was model selection conducted? Was incongruence assessed and if so, how? For all of the Bayesian analyses (MrBayes and Beast), there is no mention of how both stationarity and convergence were assessed.
R= which model do you refer to? nuclotide substitution models? We did not perform any inconsistency analysis, so cpDNA and nDNA were analyzed separately in phylogenetic analyses.
Made all minor suggestions and corrections:
Line 19: Add "they" before "serve"
Line 24: Add "analyses" after "distance"
Line 26: Change "by" to "of"
Line 55: A subject is missing from this sentence after the word "allow," perhaps include "us", "botanists", or "biologists"
Lines 66-67: The rosette is said to be less than 20 cm first, but then up to 20 cm on the next line. It cannot be both less than 20 cm and 20 cm.
Line 94: Change "between" to "among"
Line 317: The sentence reads awkwardly. I would say "The alignment was 1114 bp long and consisted of 227 nDNA sequences" or something like that.
Line 376: The grammar needs to be fixed in this sentence.
Figure 5: The image looks stretched vertically.
Line 450: Add "model" after "allopatric"
Line 461: Remove the word "Well,"
Lines 491-493: Didn't Granados Mendoza et al. 2017 Botany 95(7) do this in a group of Tillandsia? Note that I am not an author on that paper.
R= No, they do a phylogenetic analysis of Tillandsiodeae and used only one gene (matK).
